# Clinical Impact of FDG-PET/CT Compared with CE-CT in Response Monitoring of Metastatic Breast Cancer

**DOI:** 10.3390/cancers13164080

**Published:** 2021-08-13

**Authors:** Mohammad Naghavi-Behzad, Hjalte Rasmus Oltmann, Tural Asgharzadeh Alamdari, Jakob Lykke Bülow, Lasse Ljungstrøm, Poul-Erik Braad, Jon Thor Asmussen, Marianne Vogsen, Annette Raskov Kodahl, Oke Gerke, Malene Grubbe Hildebrandt

**Affiliations:** 1Department of Clinical Research, University of Southern Denmark, 5000 Odense, Denmark; Hjalte.rasmus.oltmann@rsyd.dk (H.R.O.); Tural.asgharzadeh.alamdari@rsyd.dk (T.A.A.); Jakob.lykke.bulow2@rsyd.dk (J.L.B.); Lasse.ljungstrom@rsyd.dk (L.L.); Poul-erik.braad@rsyd.dk (P.-E.B.); Marianne.vogsen@rsyd.dk (M.V.); Annette.kodahl@rsyd.dk (A.R.K.); Oke.Gerke@rsyd.dk (O.G.); Malene.Grubbe.Hildebrandt@rsyd.dk (M.G.H.); 2Department of Nuclear Medicine, Odense University Hospital, 5000 Odense, Denmark; 3Open Patient Data Explorative Network (OPEN), Odense University Hospital, 5000 Odense, Denmark; 4Centre for Personalized Response Monitoring in Oncology, Odense University Hospital, 5000 Odense, Denmark; 5Department of Radiology, Odense University Hospital, 5000 Odense, Denmark; Jon.asmussen@rsyd.dk; 6Department of Oncology, Odense University Hospital, 5000 Odense, Denmark; 7Centre for Innovative Medical Technology, Odense University Hospital, 5000 Odense, Denmark

**Keywords:** FDG-PET/CT, response monitoring, metastatic breast cancer, clinical impact

## Abstract

**Simple Summary:**

The method of treatment evaluation in patients with chronic breast cancer may affect clinical decision making and treatment protocols. In this study, we compared the two imaging modalities for the evaluation of treatment responses in 65 patients with spread breast cancer. We included 34 patients who underwent contrast-enhanced computed tomography (CE-CT) and 31 patients who underwent positron emission tomography/computed tomography (FDG-PET/CT). Then, we compared the response categories and clinical effects within the two modalities during a follow-up period of an average of 17.3 months. Our results showed that CE-CT modality reported more scans as stable disease, while FDG-PET/CT modality reported regressive metastatic disease more often. This means that FDG-PET/CT responds more precisely with respect to the changes in patients’ clinical condition, while CE-CT tends to report stable disease in most of the scans. Therefore, FDG-PET/CT may be a more suitable imaging modality than CE-CT for the evaluation of treatment in patients with metastatic breast cancer.

**Abstract:**

We compared response categories and impacts on treatment decisions for metastatic breast cancer (MBC) patients that are response-monitored with contrast-enhanced computed-tomography (CE-CT) or fluorodeoxyglucose-positron emission tomography/computed tomography (FDG-PET/CT). A comparative diagnostic study was performed on MBC patients undergoing response monitoring by CE-CT (*n* = 34) or FDG-PET/CT (*n* = 31) at the Odense University Hospital (Denmark). The responses were assessed visually and allocated into categories of complete response (CR/CMR), partial response (PR/PMR), stable disease (SD/SMD), and progressive disease (PD/PMD). Response categories, clinical impact, and positive predictive values (PPV) were compared for follow-up scans. A total of 286 CE-CT and 189 FDG-PET/CT response monitoring scans were performed. Response categories were distributed into CR (3.8%), PR (8.4%), SD (70.6%), PD (15%), and others (2.1%) by CE-CT and into CMR (22.2%), PMR (23.8%), SMD (31.2%), PMD (18.5%), and others (4.4%) by FDG-PET/CT, revealing a significant difference between the groups (*P* < 0.001). PD and PMD caused changes of treatment in 79.1% and 60%, respectively (*P* = 0.083). PPV for CE-CT and FDG-PET/CT was 0.85 (95% CI: 0.72–0.97) and 0.70 (95% CI: 0.53–0.87), respectively (*P* = 0.17). FDG-PET/CT indicated regression of disease more frequently than CE-CT, while CE-CT indicated stable disease more often. FDG-PET/CT seems to be more sensitive than CE-CT for monitoring response in metastatic breast cancer.

## 1. Introduction

Breast cancer is the most common cancer type among women in Europe and the leading cause of female cancer death in most European countries [1,2]. Despite optimized treatment for early-stage breast cancer, these patients still have a substantial risk of relapse. Metastatic breast cancer (MBC) is considered incurable and requires life-long medical treatment along with a reliable modality for the evaluation of treatment effects [3]. Longitudinal response monitoring should assess therapy effects over several treatment intervals with the intention of improving clinical decision making in MBC patients [4,5].

Contrast-enhanced computed tomography (CE-CT) and the corresponding Response Evaluation Criteria in Solid Tumors (RECIST 1.1) have been suggested for response monitoring of solid tumors [6,7]. Although most international guidelines do not provide specific recommendations on the choice of modality for response monitoring of MBC [8], the CE-CT and RECIST tend to be the most widely used methods in clinical practice and experimental trials [8,9,10].

Up to 70% of MBC patients experience bone involvement during their disease period [11], which is more common in hormone receptor-positive disease [12]. Bone lesions are hardly detected and monitored by CE-CT since active malignant lesions in the bones are difficult to distinguish from osteosclerotic recovering lesions [10,13,14]. This poses a challenge when assessing whether bone metastases are progressing, stable, or have responded to the treatment [15].

^18F^Fluorodeoxyglucose positron emission tomography with integrated computed tomography (FDG-PET/CT) has proven higher sensitivity compared to conventional imaging procedures such as CE-CT regarding the detection of distant metastasis in MBC patients [14,16]. A meta-analysis compared the sensitivity of FDG-PET/CT and CE-CT for detecting bone metastases and observed FDG-PET/CT as the superior modality with higher sensitivity (89.7% vs. 72.9%) [13]. It has also been shown that FDG-PET/CT is a better predictor of progression-free and disease-specific survival than CE-CT [17]. Hence, FDG-PET/CT has been suggested to be advantageous to CE-CT for response monitoring of MBC patients [17,18,19], but to our knowledge, no previous studies compared these methods concerning longitudinal response monitoring.

In patients with metastatic breast cancer, we aimed to explore the impact of response monitoring on treatment decisions by comparing CE-CT with FDG-PET/CT. The study objectives were to compare response categories, image-guided treatment decisions, and positive predictive values (PPV) for CE-CT and FDG-PET/CT, respectively.

## 2. Materials and Methods

This comparative diagnostic study was carried out at the Departments of Nuclear Medicine and Oncology at Odense University Hospital (Odense, Denmark) from September 2017 until June 2019. A longitudinal response assessment was retrospectively compared for MBC patients monitored primarily with CE-CT or FDG-PET/CT over several treatment intervals. The study protocol was approved by the Danish Patients’ Safety Authority (approval code: 3-3013-2448/1), and written consent forms were obtained from enrolled patients.

### 2.1. Study Design and Subjects

Women who received treatment for MBC at the Department of Oncology between September 2017 and December 2017 and had undergone CE-CT or FDG-PET/CT for treatment response monitoring were considered eligible for the study.

The inclusion criteria were biopsy-verified MBC (de novo or recurrent); baseline and at least one follow-up scan for response monitoring; use of either FDG-PET/CT or CE-CT as the main response monitoring modality; standard response monitoring protocol with regular imaging intervals (9–12 weeks on average) depending on treatment regime [20]; and regular clinical follow-up. The exclusion criteria were other known disseminated malignancy; monitoring mainly by magnetic resonance imaging (MRI) due to brain metastases; missing clinical data; and response monitoring by both modalities (CE-CT and FDG-PET/CT).

The modality used for response monitoring was generally decided by the oncologist who met the patient at the initial visit. Patients were mainly treated by the same oncologist during follow-up visits and the response monitoring modality was chosen with no internal algorithm to guide the choice of response monitoring modality. Patients were allocated arbitrarily to the treating oncologists with no distinctions due to clinical or performance status. Hence, the choice was mainly at the discretion of the oncologist based on personal preferences to choose either CE-CT or FDG-PET/CT as the response monitoring modality.

The patients were divided into the CE-CT and FDG-PET/CT groups defined by the imaging modality used for response monitoring. One scan performed on the opposite imaging modality was accepted for both groups.

The radiologists and nuclear medicine specialists made qualitative visual assessments in the clinical routine for the CE-CT and FDG-PET/CT scans, respectively. CE-CT assessments were categorized according to scan report into response categories of complete response (CR), partial response (PR), stable disease (SD), progressive disease (PD), mixed response (MR), or equivocal answer (EA). RECIST 1.1 criteria were applied when measurable lesions were present, allowing RECIST evaluation [6]. If the RECIST category differed from the visual response category, the worst prognostic response category was registered to render the response category more in line with the decision made in the clinic.

The corresponding categories were defined for qualitative visual assessments of FDG-PET/CT. The metabolic activity was taken into account as complete metabolic response (CMR), partial metabolic response (PMR), stable metabolic disease (SMD), progressive metabolic disease (PMD), mixed metabolic response (MMR), and equivocal metabolic answer (EMA). The follow-up scans were compared to the baseline and preceding scans, and the decision of multi-disciplinary radiology conference was considered in cases of uncertainty.

Response categories were subsequently dichotomized into groups of progressive and non-progressive diseases. Hence, all scans reporting PD/PMD were categorized as progressive disease, whereas all other categories were defined as non-progressive disease, the latter including MR/MMR and EA/EMA.

Clinical decisions were made by the oncologist based on the scan report, the patient’s clinical performance status, the patient’s request, and the potential toxicity of ongoing treatment. Treatment decisions were recorded for each interval and were labeled as *progression-induced change of treatment* if the oncologist decided to change the treatment due to the progression observed during imaging and/or signs of clinical progression. Reasons other than clinical progression for change of treatment, including side effects, completion of chemotherapy cycles, patients’ request, no available treatment, and complete remission, were not considered as *progression-induced change of treatment*.

### 2.2. Imaging Techniques

#### 2.2.1. FDG-PET/CT

Patients fasted for six hours before each scan. The tracer 18F-FDG was administered intravenously at a dose of 4 MBq/kg. Patients then rested and rehydrated with 800 mL of water. Scans were performed 60 min (±15 min) after 18F-FDG injection. A low-dose CT scan was performed from the skull to the proximal femora immediately followed by a PET scan of the same area. FDG-PET/CT scans were performed by using either General Electric Discovery STE, Discovery VCT, or Discovery RX (GE Medical Systems, Milwaukee, USA) with the following settings: CT-scan 140 kVp and 30–110 mA Smart mA; rotation time 0.8 sec.; pitch 1.375:1; Noise Index 25; and detector coverage 40 mm. Transverse images were reconstructed by using filtered back projection with a standard filter, slice thickness of 3.75 mm, and interval of 3.27 mm. PET scans were performed in 3D with a scan time of 2.5 min/frame. Images were reconstructed iteratively by using the GE VUE Point algorithm with 2 iterations, 21 or 28 subsets, and slice thickness of 3.3 mm. PET/CT acquisitions and reconstructions were performed in compliance with EANM guidelines, and the image quality was validated against EARL criteria [21].

#### 2.2.2. CE-CT

The diagnostic CT scans were obtained on either GE VCT, GE VCT XT, GE HD 750HD, Siemens Somatom Definition Flash, or Siemens Somatom Force. The settings for the GE scanners were as follows: 120 kV and 100–750 mA Smart mA; Auto mA; rotation time 0.5 s; pitch 0.984:1; Noise Index from 40 to 47 depending on the scanner type; and either HD 750 or VCT due to the detector specification. The ASiR level was set to 40% and detector coverage to 40 mm. Scans were generally performed over the thorax, abdomen, and pelvis in baseline and all follow-up scans. There were three reconstructions made with Soft, Stnd., and Lung algorithm. Soft was used for 0.625 mm axial slices and 5 mm coronal, and Sagittal. Stnd. and Lung was reconstructed in 5 mm axial slices. The settings for the Siemens Flash scanner were as follows: 120 kV; ref mas 150; rotation time 0.5 s; and pitch 0.9. The detector coverage was 40 mm 0.6 mm × 128, SAFIRE level 3. The settings for the Siemens FORCE scanner were as follows: 120 kV; ref mas 110; rotation time 0.5 s; and pitch 0.6. The detector coverage was 60 mm 0.6 mm × 192, ADMIRE level 2. Reconstructions for the Flash scanner were made with I31f medium Smooth and I50f medium Sharp ASA kernels. I31f was used for 0.625 mm and 5 mm axial, coronal, and sagittal slices. The lung window was reconstructed with I50f medium Sharp ASA in 5 mm axial slices. Reconstructions for the Force scanner are reconstructions made with Br40 and Bl57 kernels. Br40 was used for 0.625 mm and 5 mm axial, coronal, and sagittal slices. The lung window was reconstructed with Bl57 in 5 mm axial slices. The contrast enhancement scheme for diagnostic CT was generally contrast medium Optiray 300 (1 mL/kg on patients’ weight) and the scan started 70 s after the injection. Flow rate was usually 3 mL/s, while reduced by radiographer’s discretion in the case of a fragile vein. Antecubital vein was preferred, unless the patient already had other peripheral venous access. Iomeron or Omnipaque were administered in similar doses in patients with previous allergic reactions [6].

### 2.3. Data Collection and Variables

Data were extracted from medical records and scan reports. Extracted data included age, performance status [22], and clinical and histopathological data. For patients with more than one primary breast cancer, we used the data for the primary cancer that had most likely resulted in metastasis (i.e., had the same molecular profile as metastasis). In a few patients with de novo metastatic cancer, information from the primary breast biopsy was used since biopsy was not obtained from the metastatic lesion due to local procedure.

### 2.4. Outcome Measures and Statistical Analyses

Study groups were checked for comparability regarding baseline characteristics. Response categories were compared between the study groups. The response monitoring period was defined as the time interval between the first treatment line and the last available scan. A reference standard for true and false progression was assessed using information from subsequent follow-up scans. Hence, true progression could be applied in two situations: (1) when progression (PD/PMD) caused treatment change, and the subsequent scan revealed regression (PR/PMR or CR/CMR); and (2) when progression (PD/PMD) did not results in treatment change, and the subsequent scan revealed further progression (PD/PMD). False progression could be applied when progression (PD/PMD) did not result in treatment change, and the subsequent scan revealed non-progressive disease (SD/SMD, PR/PMR, or CR/CMR). Then, the positive predictive value (PPV) could be compared for the CE-CT and FDG-PET/CT groups, respectively. The PPV was calculated as $\frac{\mathrm{True}\;\mathrm{progression}}{\mathrm{True}\;\mathrm{progression}\;+\;\mathrm{False}\;\mathrm{progression}}$. The PPVs for CE-CT and FDG-PET/CT were supplemented by Wilson-score-based 95% confidence intervals (95% CIs). A chi-squared test was used to test for differences in distributions between groups. All data were analyzed by using STATA/IC software (version 15.0, StataCorp, College Station, TX, USA). *P*-values less than 0.05 were considered statistically significant.

## 3. Results

### 3.1. Baseline Characteristics

A total of 109 patients were eligible for the study, while 44 patients were subsequently excluded for different reasons, as shown in Figure 1. Of the 65 included patients who were response monitored between 2009 and 2017, 34 were enrolled in the CE-CT group and 31 in the FDG-PET/CT group. Clinical and histopathological characteristics of the primary tumor and metastatic disease are summarized in Table 1 and Table 2, respectively. The study groups were comparable regarding most of the baseline characteristics.

### 3.2. Treatment Protocol and Metastatic Site

Exposure to different treatment types and metastatic sites reported during the follow-up period are reflected in Table 3. Received treatments were comparable between the study groups without any significant difference. However, CE-CT reported metastatic bone (58% vs. 47.6%; *P* = 0.022) and liver (19.2% vs. 7.9%; *P* = 0.001) metastases significantly more frequently than FDG-PET/CT during response monitoring scans.

### 3.3. Response Monitoring Scans

A total of 333 CE-CT scans and 230 FDG-PET/CT scans, including baseline and response monitoring scans, were performed in 65 patients with median response monitoring periods of 23 (range: 6–90) and 11 (2–103) months for patients monitored with CE-CT and FDG-PET/CT, respectively (*P* = 0.007). FDG-PET/CT was performed once in 10 out of 34 (29.4%) patients in the CE-CT group, while CE-CT was performed once in 13 out of 31 (41.9%) patients in the FDG-PET/CT group. An illustration of response evaluation intervals with response categories for each patient is shown in Figure 2. The figure illustrates that SD (yellow color) is highly represented in the CE-CT group, while PMR (green color) is highly represented in the FDG-PET/CT group.

### 3.4. Response Categories, Clinical Impact, and Positive Predictive Value

A total of 286 CE-CT and 189 FDG-PET/CT response monitoring scans were performed. RECIST was applied in the response evaluation of 178 out of 286 (62.2%) CE-CT scans and by visual assessment in the remaining scans (37.8%), while response evaluation was performed by visual assessment in all FDG-PET/CT scans. The decision on response categories was different between RECIST and visual assessment in 13 CE-CT scans; however, the difference was balanced regarding the report of worse response category by either of them.

Within the response monitoring scans, 15.4% (44/286) and 11.6% (22/189) of scans followed by *progression-induced change of treatment* occurred in CE-CT group and FDG-PET/CT group, respectively. Time to the detection of first progression leading *progression-induced treatment change* was 9.3 months shorter in the FDG-PET/CT group (15 patients) with a median of 8.8 (3.5–61.3) months compared to the CE-CT group (20 patients) with a median of 18.1 (2.8–54.9) months (*P* = 0.04). Response categories and treatment decisions followed by CE-CT and FDG-PET/CT scans have been summarized in Table 4. A statistically significant difference in the distribution of response categories was observed between the groups (*P* < 0.001). FDG-PET/CT reported regressive disease more frequently (46.0%) than CE-CT (12.2%) did, while stable disease was reported more often in CE-CT (70.6%) compared with FDG-PET/CT (31.2%). The PPV was 0.85 (95% CI: 0.72–0.97) and 0.70 (95% CI: 0.53–0.87) for CE-CT and FDG-PET/CT, respectively (*P* = 0.17). Additionally, when CE-CT reported PD, treatment was changed in 79.1% (34/43), and treatment was changed in 60% (21/35) (*P* = 0.083) when FDG-PET/CT reported PMD. By observing the patients who had a *progression-induced change of treatment*, progressive disease was reported in 95.5% scans by FDG-PET/CT and 77.3% by CE-CT (*P* = 0.08).

## 4. Discussion

In the current study, FDG-PET/CT reported regressive disease more frequently than CE-CT (46.0% vs. 12.2%) in patients with metastatic breast cancer undergoing longitudinal response monitoring, while CE-CT deemed stable disease more often (70.6% vs. 31.2%) than FDG-PET/CT (Figure 2). Progression was relatively equally reported by CE-CT and FDG-PET/CT (15% and 18.5%, respectively), and no statistically significant difference was observed for an approached positive predictive value for progression between the groups. Since treatment change would typically occur when progression is deemed and progression was quite equally reported, our findings indicate that treatment strategy may not change significantly when choosing either CE-CT or FDG-PET/CT as response monitoring modality. On average, the first progression resulting in treatment change in the clinic (*progression-induced treatment change*) was detected 9.3 months earlier (*P* = 0.04) in patients response monitored by FDG-PET/CT (15 patients) compared to patients response monitored by CE-CT (20 patients); however, we cannot make firm conclusions due to the lack of sample size and short follow-up time in this sub-group of patients. However, FDG-PET/CT may have the potential to detect progression earlier than CE-CT due to the hypothesis that change in cancer activity would occur before the presentation of morphological changes.

Clinicians were more likely to change treatment when CE-CT deemed progression than when FDG-PET/CT deemed progression (79.1% vs. 60%), while the reporting of progressive disease was more in line with the decision made in the clinic (progression-induced change of treatment) in the FDG-PET/CT group than in the CE-CT group (95.5% vs. 77.3%, *P* = 0.08).

Patients included in this study were biopsy-verified MBC patients in an unselected, unique consecutive cohort monitored with primarily one of the two compared imaging modalities. They were, apparently, by part randomly allocated to monitoring with FDG-PET/CT or CE-CT, and hence the two groups were comparable concerning most characteristics (Table 1 and Table 2). However, the patients monitored by CE-CT had a significantly longer follow-up time. This could be explained by the fact that FDG-PET/CT is a more novel option for treatment monitoring than CE-CT, and longer follow-up time for patients monitored by FDG-PET/CT would have been preferable. Clinically relevant data and clinical decision-making data were extracted in order to evaluate the clinical impact of each modality. A considerable large quantity of scans for both modalities was included, and response categories were interpreted by the scan reports representative of routine clinical practice.

The limitations of the study were the single-centre observational design, low sample size, and heterogeneity of the included patients such as a different number of response monitoring series, treatment protocols, time point of diagnosis for metastatic disease (2015 vs. 2016), and follow-up period. Lack of a reference standard for confirmation of true progression/regression was a limitation since we followed the decision made in the clinic with the potential of reporting false positive “true progression”. Moreover, we allowed one opposite scan type to be performed in both groups, which may have blurred our findings to some degree. Additionally, the RECIST criteria were applied in 62% of CE-CT scans, and no standardized response evaluation criteria, such as the PET Response Evaluation Criteria in Solid Tumors (PERCIST), were applied for FDG-PET/CT. Furthermore, we have not categorized response evaluations based on different treatments and histopathology profiles, which could impact our results [23].

To our knowledge, this is the first study that compared the impact of response categorization and clinical decision making of using different scan modalities over several treatment intervals in MBC patients. Previously, in a study by Riedl et al., it has been shown that FDG-PET/CT had a higher predictive value than CE-CT for long-term survival when the PERCIST and RECIST were applied [17]. According to their results, CE-CT tended to report SD more often, while FDG-PET/CT reported PMD more often. These patterns are in line with the findings of our study to some degree and could be explained by the inability of CE-CT to distinguish osteoblastic bone lesions from bone healing. They also showed that FDG-PET/CT could be a better predictor for progression-free survival and disease-specific survival compared to CE-CT [17], which was proposed in some other studies as well [24,25]. These results indicate that FDG-PET/CT may be superior to CE-CT in the response assessment of MBC patients [17,24,25].

We observed that clinicians tended to change treatments more often based on progression reported by CE-CT than by FDG-PET/CT. This could be explained by a lack of evidence in this field and the absence of FDG-PET/CT in international and national guidelines as a modality of choice for the response evaluation of MBC patients [26]. Liver metastases were reported more frequently in CE-CT response monitoring scans, which could reflect the lower specificity of CE-CT for the diagnosis of liver metastases compared to FDG-PET/CT [27].

With a major difference in reporting more regressive disease by FDG-PET/CT (23.8% vs. 8.4%) and more stable disease by CE-CT (70.6% vs. 31.2%), it appears that FDG-PET/CT could have the potential to provide an early indication of treatment effect for different treatment options in this group of patients. This may provide the advantage of the knowledge that the treatment has a positive effect on cancer and, thereby, confidence for the clinician and decision-makers of treatment benefit as it may increase the quality of life and treatment motivation for the patients [5,28], and it may have a potential for improving assessment of treatment efficacy in clinical trials.

Since our results were part of the clinical routine, a standardized set of response criteria such as PERCIST was not applied for FDG-PET/CT. However, we showed in previous studies that the clinical application of PERCIST is feasible [29] and has the potential to increase the level of inter-observer agreement and reliability when compared with qualitative visual assessment [30].

Targeted treatment methods, such as anti-HER2 therapy, have been used for a long time in breast cancer resulting in improved survival [31,32]. These treatment lines may require more accurate response monitoring methods to serve as surrogates for survival, to keep patients in the most effective treatment tract over time, and to ensure the cost-effectiveness of modern treatments. Along with the increasing use of molecular directed treatments, we should encourage a move from changes in tumor size towards changes in cancer metabolism when evaluating therapeutic responses [5,33]. Emerging supplemental information is coming from the area of liquid biopsies that could provide more detailed information about active cancer cells and, therefore, be more specific in treatment monitoring [34,35].

The efficacy of using FDG-PET/CT as response monitoring modality may vary for different histopathological profiles of metastases [23]. This restriction refers to the limited performance of FDG-PET/CT due to lower FDG-uptake in patients with invasive lobular carcinoma histotype [36,37].

On the other hand, whole body MRI revealed a comparable diagnostic accuracy to PET/CT regarding the detection of bone metastasis [13,38], and it has been shown that whole body MRI could detect the progression earlier than CE-CT in patients with bone-only MBC [39]. Therefore, whole body MRI could be considered as a potential alternative strategy in the response monitoring of MBC patients, while future prospective studies are needed to compare the clinical impact of all three modalities through comparable groups.

Since optional imaging modalities for response evaluation are suggested by current international guidelines [8], this means that opposing methods might be used in different institutions. Our data suggested that imaging modalities may provide varying information about response to treatment, but it could not show whether any of the modalities detected the progression, resulting in treatment change earlier than the other. The hypothesis that using FDG-PET/CT for monitoring response has an impact on treatment strategy is, therefore, a perspective for future research. Prospective studies with longer follow-up time, applying semi-quantitative response assessments such as the PERCIST criteria to FDG-PET/CT on patients receiving similar treatment types, could potentially result in a better understanding of the clinical impact of FDG-PET/CT in the response monitoring of MBC patients. It would also be valuable to compare the long-term survival and the costs of response monitoring within the modalities along with the clinical impact. The high frequency of bone involvement in these patients is still a strong indicator for further research into the role of FDG-PET/CT in response monitoring of MBC to provide better evidence-based recommendations.

## 5. Conclusions

In response monitoring of patients with metastatic breast cancer, regressive disease was reported more frequently by FDG-PET/CT, and stable disease was reported more often by CE-CT. Hence, this study indicates that FDG-PET/CT may be more sensitive for confirming treatment effects than the conventional CE-CT. Progressive disease was deemed quite equally, and positive predictive values were quite similar for the two modalities; however, clinicians tended to be more likely to change treatment when CE-CT suggested progression than when FDG-PET/CT suggested progression. Time to detection of treatment failure may impact treatment decisions and should be analyzed for the two modalities in future studies with larger sample size and prospective study design.

## Figures and Tables

**Figure 1 cancers-13-04080-f001:**
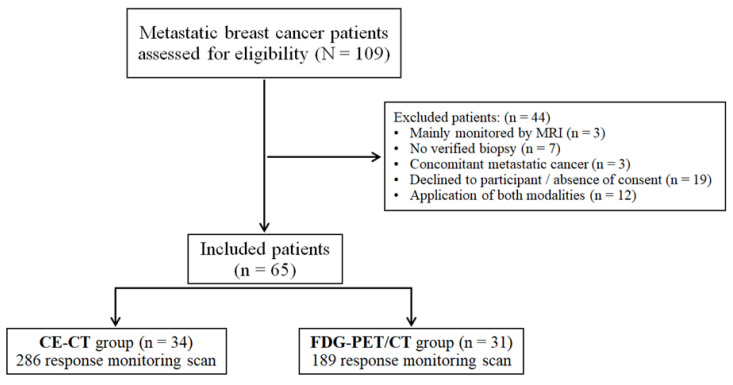
Patient flowchart and distribution of the included patients to study groups (CE-CT: contrast-enhanced computed tomography; FDG-PET/CT: Fluorodeoxyglucose positron emission tomography with integrated computed tomography; MRI: magnetic resonance imaging).

**Figure 2 cancers-13-04080-f002:**
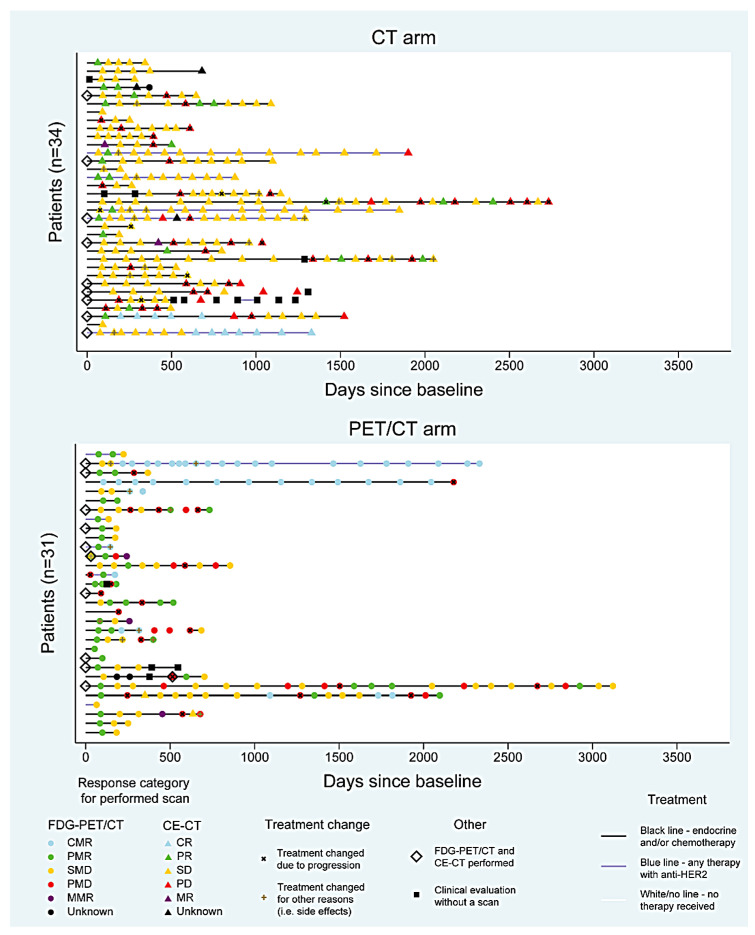
Visualization of response categories and treatment changes for study groups (CE-CT: contrast-enhanced computed tomography; FDG-PET/CT: fluorodeoxyglucose positron emission tomography with integrated computed tomography; CR: complete response; PR: partial response; SD: stable disease; PD: progressive disease; MR: mixed response; CMR: complete metabolic response; PMR: partial metabolic response; SMD: stable metabolic disease; PMD: progressive metabolic disease; MMR: mixed metabolic response).

**Table 1 cancers-13-04080-t001:** Clinicopathological characteristics of primary breast cancer.

Characteristics ^a^		Study Groups		*P-*Value
		CE-CT	PET/CT	
**Primary tumor size** (mm)		18 (3–65)	20 (1–80)	0.85
**Bilateral cancer**		2 (5.9)	5 (16.1)	0.24
**Histopathology**	Ductal	26 (76.5)	23 (74.2)	0.59
	Lobular	5 (14.7)	3 (9.7)	
	Adenocarcinoma	3 (8.8)	3 (9.7)	
	Unknown	0 (0)	2 (6.5)	
**Primary surgery (lumpectomy/mastectomy)**		23 (67.6)	27 (87.1)	0.06
**Estrogen receptor status**	Positive	31 (91.2)	25 (80.7)	0.43
	Negative	2 (5.9)	3 (9.7)	
	Unknown	1 (2.9)	3 (9.7)	
**HER-2 status**	Positive	5 (14.7)	6 (19.4)	0.94
	Negative	21 (61.8)	18 (58.1)	
	Unknown	8 (23.5)	7 (22.6)	
**Tumor grade**	Grade 1	6 (17.7)	6 (19.4)	0.98
	Grade 2	14 (41.2)	11 (35.5)	
	Grade 3	7 (20.6)	8 (25.8)	
	Unknown	7 (20.6)	6 (19.4)	
**Ki-67 proliferation** (%)		20 (1–80)	50 (10–95)	0.16
**Lymph node involvement**	None	7 (20.6)	6 (19.4)	0.71
	Micro-metastasis	4 (11.8)	3 (9.7)	
	Macro-metastasis	13 (38.2)	16 (51.6)	
	Unknown	10 (29.4)	6 (19.4)	
**Treatment protocol**	Neo-adjuvant treatment	2 (5.88)	8 (25.8)	0.09
	Endocrine treatment	23 (67.7)	17 (54.8)	
	No treatment / unknown	9 (26.5)	6 (19.4)	
**Radiotherapy (breast/breast + axilla)**		20 (58.8)	21 (67.7)	0.31

CE-CT: contrast-enhanced computed tomography; FDG-PET/CT: fluorodeoxyglucose positron emission tomography with integrated computed-tomography; HER-2: human epidermal growth factor receptor-2. ^a^ Data were shown as median (interquartile range) or frequency (percentage).

**Table 2 cancers-13-04080-t002:** Clinicopathological characteristics of metastatic disease ^a^.

Characteristics		Study Groups		*P-*Value
		CE-CT	PET/CT	
**Year of diagnosis**		2015 (2010–2017)	2016 (2009–2017)	0.02
**Age at diagnosis** (year)		67.0 (31.0–84.5)	63.4 (32.9–85.8)	0.69
**Performance status**	0	12 (35.3)	13 (41.9)	0.36
	1	10 (29.4)	13 (41.9)	
	≥2	3 (8.8)	1 (3.2)	
	Unknown	9 (26.5)	4 (12.9)	
**Time until relapse ^b^** (months)		78.0 (0–271.4)	81.8 (0–307.5)	0.71
**Histopathology**	Ductal	4 (11.8)	5 (16.1)	0.86
	Lobular	4 (11.8)	2 (6.5)	
	Adenocarcinoma	18 (52.9)	18 (58.1)	
	Unknown	8 (23.5)	6 (19.4)	
**De novo metastatic cancer**		8 (25.5)	4 (12.9)	0.35
**Estrogen receptor status**	Positive	30 (88.2)	28 (90.3)	0.50
	Negative	2 (5.9)	3 (9.7)	
	Unknown	2 (5.9)	0 (0)	
**HER-2 status**	Positive	5 (14.7)	6 (19.4)	0.46
	Negative	22 (64.7)	22 (71.0)	
	Unknown	7 (20.6)	3 (9.7)	
**Origin of biopsy**	Bone	5 (14.7)	13 (41.9)	0.07
	Liver	6 (17.7)	3 (9.7)	
	Lung/Pleural fluid	8 (23.5)	7 (22.6)	
	Breast/lymph nodes	15 (44.1)	8 (25.7)	
**Region of metastases at baseline scan**	Bone-only metastasis	4 (11.8)	4 (12.9)	0.89
	Bone	22 (64.7)	21 (67.7)	0.80
	Liver	8 (25.5)	8 (25.8)	0.83
	Lung	10 (29.4)	11 (35.5)	0.60
	Regional lymph nodes	12 (35.3)	9 (29.0)	0.59
	Distant lymph nodes	18 (52.9)	17 (54.8)	0.88
	Pleura/pleural effusion	3 (8.8)	6 (19.4)	0.22
	Breast/local recurrence	7 (20.6)	6 (19.4)	0.90
	Soft tissue	1 (2.9)	4 (12.9)	0.13
	Others ^c^	2 (5.9)	3 (9.7)	0.57

CE-CT: contrast-enhanced computed tomography; FDG-PET/CT: fluorodeoxyglucose positron emission tomography with integrated computed-tomography, HER-2: human epidermal growth factor receptor-2. ^a^ Data shown as median (interquartile range) and frequency (%). ^b^ Time until relapse for patients with primary disseminated disease was considered zero. ^c^ Others comprised ascites, adrenal glands, thyroid, uterus, colon, and skin.

**Table 3 cancers-13-04080-t003:** Treatment regimens and detected metastasis during follow-up.

Characteristics		Study Groups	
		CE-CT (286 scan)	FDG-PET/CT (189 scan)
**Received treatments during follow-up ^a^**	Endocrine therapy	29 (85.3)	25 (80.6)
	Bone-targeted therapies	24 (70.6)	23 (74.2)
	Chemotherapy	22 (64.7)	18 (58.1)
	CDK4/6 inhibitors	19 (55.9)	15 (48.4)
	Anti-HER2 therapy	5 (14.7)	6 (19.4)
	Palliative radiotherapy	5 (14.7)	1 (3.2)
**Metastatic sites during follow-up ^b^**	Bone	166 (58.0)	90 (47.6)
	Liver	55 (19.2)	15 (7.9)
	Lung/plural	110 (38.5)	66 (34.9)
	Regional/distant lymph nodes	112 (39.2)	67 (35.4)
	Others ^c^	48 (16.8)	28 (14.8)

CE-CT: contrast-enhanced computed tomography, FDG-PET/CT: fluorodeoxyglucose positron emission tomography with integrated computed-tomography. ^a^ Data shown as frequency (percentage) out of patients’ number. ^b^ Data shown as frequency (percentage) out of number of scans. ^c^ Others comprised ascites, adrenal glands, thyroid, uterus, colon, soft tissue, and skin.

**Table 4 cancers-13-04080-t004:** Response categories for patients monitored by CE-CT and FDG-PET/CT .

		Treatment Actions	Response Categories ^a^					
Study Groups			CR/CMR	PR/PMR	SD/SMD	PD/PMD	MR/MMR	EA/EMA
**CE-CT**	Scans followed by progression-induced treatment change ^b^ (*n* = 44)		0 (0)	1 (2.3)	8 (18.2)	34 (77.3)	0 (0)	1 (2.3)
	Total scans ^c^ (*N* = 286)		11 (3.8)	24 (8.4)	202 (70.6)	43 (15)	2 (0.7)	4 (1.4)
**PET/CT**	Scans followe by progression-induced treatment change ^b^ (*n* = 22)		0 (0)	0 (0)	0 (0)	21 (95.5)	1 (4.5)	0 (0)
	Total scans ^c^ (*N* = 189)		42 (22.2)	45 (23.8)	59 (31.2)	35 (18.5)	4 (2.1)	4 (2.1)

CE-CT: contrast-enhanced computed tomography; FDG-PET/CT: fluorodeoxyglucose positron emission tomography with integrated computed-tomography; CR: complete response; PR: partial response; SD: stable disease; PD: progressive disease; MR: mixed response; EA: equivocal answer; CMR: complete metabolic response; PMR: partial metabolic response; SMD: stable metabolic disease; PMD: progressive metabolic disease; MMR: mixed metabolic response; EMA: equivocal metabolic answer. ^a^ Data have been shown as frequency (percentage) of each line. ^b^ Progression-induced treatment change defined as progression observed in imaging and/or clinical progression resulting in treatment change. ^c^ Scans from opposite modality were excluded in both study groups.

## Data Availability

Anonymous data as well as statistical methods could be shared upon reasonable request to the corresponding author.

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
