# Peer review of "Clinical Impact of FDG-PET/CT Compared with CE-CT in Response Monitoring of Metastatic Breast Cancer"

_cancers, 2021, doi:10.3390/cancers13164080_

Round 1

Reviewer 1 Report

Dear authors,

I have no further comments.

Reviewer 2 Report

The authors re-submit this manuscript and had addressed all of my concerns raised in the previous submission. 

This manuscript is a resubmission of an earlier submission. The following is a list of the peer review reports and author responses from that submission.

Round 1

Reviewer 1 Report

I have read the manuscript with great interest. The clinical data for deciding the diagnosis of MBC is a very promising finding. Overall, the manuscript is well organized to meet the requirements of acceptance. However, there is minor confusion that needs to clarify in revision. It including,

  1. The ‘xx months’ will not clear for most of the readers in the Simple Summary section. It should be clarified.
  2. The ‘longitudinal response monitoring’ term should be described with a supporting description in the Introduction section.
  3. The format and position of Table 1, Table 2, and Table 3 need to correct.
  4. The authors could cite the results (For example, Figure 1 or Table 1) in the Discussion section. It will help the reader to correlate the Results with the Discussion.

Reviewer 2 Report

Dear authors, thank you for submitting your manuscript, entitled: “Clinical impact of FDG-PET/CT compared with CE-CT in response monitoring of metastatic breast cancer”.

Here some revisions in order to improve your manuscript:

-line 25: “Then, we compared the response categories and clinical effect within the two  modalities during a follow-up period of on average xx months.” Please specify the number of months.

-how patients were assigned to one imaging type or another one should be better described in the methods.

-a mention to whole-body MRI as potential alternative strategy should be mentioned.

-3.2. “Treatment protocl and metastatic site” correct the typo. There are a lot of typos like this one, please revise all the article and correct them.

-please, expand the discussion related to impact on survival outcomes (PFS, OS).

-please, expand the section related to limitation/biases of the study (e.g. small sample size, discrepancies in fup time, stratifications, reproducibility e.g. PERCIST);

Reviewer 3 Report

Dear authors,

In the present large study, you present data on the clinical impact of FDG-PET/CT compared with CE-CT regarding response monitoring in metastatic breast cancer patients.

I have the following comments:

- Your main thesis is that PET/CT indicates regression more often than CE-CT (which is more often categorized as “stable disease”). You conclude that PET/CT is thus more sensitive to therapy response. Your conclusion is essentially similar to the manuscript published by Riedl et al. which you summarize as “Their results indicate that response assessment by FDG-PET/CT may be superior to CE-CT.” (ll. 422 f.). The similarity in results nicely underlines the value of FDG-PET/CTs but necessitates some clarification regarding the novelty of your work compared to the previous study. While I understand that you have included more imaging, Riedl’s study has the advantage of having performed both imaging modalities in the same patients and having performed outcome correlations to determine which modality most closely reflects the prognostic situation.

It also seems that Riedl found the exact same patterns as you did: Any response (CR / PR) vs. no response (SD + PD) was 40:25 in PET-CT and 22:43 in CE-CT, indicating that in this study, too, PET-CT was more likely to indicate response than CE-CT (see Table 3 of the Riedl study). The study even previews that bone lesions may wrongly be categorized as PD on CE-CT, a finding exactly mirrored in your study (l. 291). Given these similarities, it would be immensely important to very clearly discern new aspects in your investigation.

- You allowed one scan to be from the opposite imaging modality. How did you count the results from this scan (e.g. a PET/CT scan obtained in the CT group)? Did you discount the findings from the scan?

- The use of the positive predictive value remains somewhat unclear. What did you base your definition on? Based on your description, a case of PD with subsequent therapy change and another PD finding on the next scan would not have met your definition of “true progression”. It is also debatable whether PD, treatment change and regression on the next scan really constitutes “true progression” in the interval. It may or may not have been progression.

Minor points:

- The follow-up in your lay abstract is still marked as “xx”.

- I would argue that Riedl’s study, as discussed above, is also a longitudinal study given that two timepoints (pre-therapy and after three months) were included and interval changes were compared. Hence, I would suggest to be cautious regarding the claim of being the “first longitudinal study” (ll. 77f.)

- Did you compare the time of diagnosis for the CT and the PET/CT groups? Were patients in the CT group more likely to be included earlier? This would explain the longer follow-up time in the CT group, but might also be a potential confounder given the time-changing treatment algorithms.

- In table 4, there is one “progression-induced treatment change” in a patient with PR. This seems surprising and, given the absence of progression, probably not “progression-induced”.

- Based on your table 4, line 358 should read 77.3%, not 73.3%, if I understand your table and the meaning of your phrase correctly.

Reviewer 4 Report

It is an interesting study aimed to compare response categories and impact on treatment decisions for metastatic breast cancer (MBC) patients response-monitored with contrast-enhanced computed-tomography (CE-CT) or FDG-PET/CT. Overall, I have concerns that the design of this study and results may not support  the conclusions. Evaluation of response in a population with both CE-CT and PET/CT may be better to address the questions and conclude PET/CT is more sensitive than CE-CT for response monitoring. My specific comments are as follows:

Page 3, Line 103-104: It would be nice to describe discretions of the oncologists to decide the modality used for response monitoring here.

Page 3, Line 109: What were the indications of oncologosts to perform the oppsite imaging modality? Would these patients had suspected lesions on CT scans and further evaluated by PET/CT? Otherwise, the PET/CT has higher sensitivity and specificty for detecting recurrences or evaluating reponses. The authors should also need to explain patients in the PET/CT arm to have CT scans evaluations to monitor response.

Page 3, Line 111-112: What's the range of your CE-CT for evaluating response for these patients? Were these CE-CT scans whole body or only brain, chest, abdomen, pelvic, or others based on the monitoring sites? Did these patients in the CE-CT arm have identical range of range of field? Otherwise, were there newly recurrent lesions found in these patients?

Table 2: There were many different sites of metastases in these patients. In each patient, were these lesions in each patients had consistent response to treatment? For instance, in a patient with lung, vertebral bone, pelvic bone, and lymph nodes metastases, were the responses of these lesions to treatments similar? If there were inconsistent responses, how would the authors to define their responses?

In the 3.4. Response categories, clinical impact, and positive predictive value,
The time to detect progression and time to progression-induced change of treatment should be presented in this section.